# Systematic Study of Effective Hydrothermal Synthesis to Fabricate Nb-Incorporated TiO_2_ for Oxygen Reduction Reaction

**DOI:** 10.3390/ma15051633

**Published:** 2022-02-22

**Authors:** So Yoon Lee, Daiki Numata, Ai Serizawa, Koudai Sasaki, Kaito Fukushima, Xiulan Hu, Takahiro Ishizaki

**Affiliations:** 1Department of Materials Science and Engineering, College of Engineering, Shibaura Institute of Technology, 3-7-5- Toyosu, Koto-ku, Tokyo 135-8548, Japan; soyoon@shibaura-it.ac.jp (S.Y.L.); ishizaki@shibaura-it.ac.jp (T.I.); 2Graduate School of Engineering and Science, Shibaura Institute of Technology, 3-7-5- Toyosu, Koto-ku, Tokyo 135-8548, Japan; mb19029@shibaura-it.ac.jp (D.N.); ac18046@shibaura-it.ac.jp (K.S.); mb21045@shibaura-it.ac.jp (K.F.); 3College of Materials Science and Engineering, Nanjing Tech University, Nanjing 211816, China; whoxiulan@njtech.edu.cn

**Keywords:** Nb doping, TiO_2_, hydrothermal synthetic process, electrocatalytic activity, ORR

## Abstract

Fuel cells are expected to serve as next-generation energy conversion devices owing to their high energy density, high power, and long life performance. The oxygen reduction reaction (ORR) is important for determining the performance of fuel cells; therefore, using catalysts to promote the ORR is essential for realizing the practical applications of fuel cells. Herein, we propose Nb-incorporated TiO_2_ as a suitable alternative to conventional Pt-based catalysts, because Nb doping has been reported to improve the conductivity and electron transfer number of TiO_2_. In addition, Nb-incorporated TiO_2_ can induce the electrocatalytic activity for the ORR. In this paper, we report the synthesis method for Nb-incorporated TiO_2_ through a hydrothermal process with and without additional load pressures. The electrocatalytic activity of the synthesized samples for the ORR was also demonstrated. In this process, the samples obtained under various load pressures exceeding the saturated vapor pressure featured a high content of Nb and crystalline TiNb_2_O_7_, resulting in an ellipsoidal morphology. X-ray diffraction results also revealed that, on increasing the Nb doping amounts, the diffraction peak of the anatase TiO_2_ shifted to a lower angle and the full width at half maximum decreased. This implies that the Ti atom is exchanged with the Nb atom during this process, resulting in a decrease in TiO_2_ crystallinity. At a doping level of 10%, Nb-incorporated TiO_2_ exhibited the best electrocatalytic activity in terms of the oxygen reduction current (*i*_ORR_) and onset potential for the ORR (*E*_ORR_); this suggests that 10% Nb-doped samples have the potential for enhancing electrocatalytic activity.

## 1. Introduction

The increasing use of mobile devices and the advances in electric vehicles have resulted in an increased demand for highly efficient energy conversion and storage devices. Although Li-ion batteries have been widely used in electronic devices over recent years, there is limited scope for improving their energy density as they approach the theoretical limit [1,2]. Thus, to satisfy the demands for enhanced energy density, power, and performance in batteries, the development of next-generation batteries and energy conversion devices is crucial. To this end, fuel cells, which are expected to possess a significantly higher energy density than currently available rechargeable batteries, have been extensively studied [3,4,5]. The oxygen reduction reaction (ORR) is important for determining the performance of fuel cells; therefore, the use of catalysts to promote ORR is essential for realizing the practical applications of fuel cells [4,5,6,7]. With regard to the materials, generally, Pt-based catalysts exhibit remarkable electrocatalytic activity for the ORR [8,9,10]; however, they suffer from disadvantages in terms of commercialization, such as high costs and long-term instability [11,12,13]. Therefore, alternative materials similar to Pt catalysts, including metal oxides [14,15], metal chalcogenides [16], and carbon materials [17,18], have been proposed.

With regard to the synthetic route, with the advent of industrialization, environmental pollution has emerged as a major problem. Thus, many researchers have focused on investigating environmental remediation technologies and environmentally friendly materials [19,20]. Among various such materials, titanium dioxide (TiO_2_) has widely been employed in photocatalytic applications owing to its outstanding performance, abundance, low cost, high stability, and environment-friendliness [21,22,23]; however, its low conductivity and reactivity are major concerns limiting its usage. By contrast, niobium (Nb) doping has been widely used to improve conductivity and electron transfer numbers by synthesizing Ti_4_O_7_. These synthesized products have exhibited high conductivity, a special crystalline structure, and good absorption properties [24,25,26,27]. The merits of using this synthesized product are as follows: First, Nb doping induces the structural evolution of TiO_2_ from the rutile phase to the anatase phase and improves its photocatalytic activity [28,29]. Second, the doping of Nb^5+^ with TiO_2_ introduces additional electrons in the conduction band. Consequently, these excess electrons can assist the rapid initial reaction of the organic decomposition process [30,31].

Based on this concept, Nb doping in TiO_2_ using an environmentally friendly method is an interesting research topic for the further development of advanced ORR catalysts. Recently, the hydrothermal synthesis (HS) method for Nb doping has emerged as a promising choice for developing efficient ORR catalysts. The HS process utilizes the high reactivity of high-temperature and high-pressure water (subcritical water) via autoclaving; this process promotes the liquid-phase synthesis of substances that are sparingly soluble in water. The relationship between the Nb doping level and HS route for enhancing the electrocatalytic activity for the ORR still remains unclear, and further studies are required. A previous study reported that doping 10% Nb in TiO_2_ could potentially promote the ORR; this information was considered as the base condition in this study for initiating the experiment [31]. Herein, we systematically demonstrate the relationship between the electrocatalytic activity for the ORR and the Nb content, by synthesizing Nb-incorporated TiO_2_ through a hydrothermal process with and without load pressures. Through this process, the samples obtained using various load pressures possessed high contents of Nb and crystalline TiNb_2_O_7_, resulting in an ellipsoidal morphology. In addition, X-ray diffraction results showed that when the Nb doping amounts were increased, the diffraction peak of anatase TiO_2_ shifted to a lower angle and the full width at half maximum decreased. This implies that the Ti atom was exchanged with the Nb atom during the process; therefore, the TiO_2_ crystallinity decreased. Furthermore, 10% Nb-incorporated TiO_2_ showed the best electrocatalytic activity in terms of the oxygen reduction current (*i*_ORR_) and onset potential for the ORR (*E*_ORR_), thereby suggesting that the 10% Nb-doped samples have the potential for enhancing electrocatalytic activity.

## 2. Materials and Methods

### 2.1. Preparation of Nb-Incorporated TiO_2_

To synthesize Nb-incorporated TiO_2_, TiOSO_4_ powder (Technical, Sigma-Aldrich, Tokyo, Japan) was used as the Ti source. First, 1.271 g of TiOSO_4_ powder was dissolved in 100 mL of pure H_2_O. To incorporate Nb into TiO_2_, the Nb doping level was tuned from 10% to 20% against the total number of Ti atoms using NbCl_5_ powder (99+%-Nb, Strem Chemicals, Tokyo, Japan) and dissolving it in a solution containing TiOSO_4_ powder. To monodisperse the prepared mixture, ultrasonication was performed. To adjust the pH of the prepared mixture, a 28% ammonia solution was used to set the pH to 10.

The following procedure was performed for HS: First, the prepared mixture, which was tuned to pH = 10, was input to a 100 mL autoclave container. The high-pressure mechanism of the high-pressure machine is designed to fill the inside of the pipe with high-pressure water using pure water as a mediator. Because of this, the sample filling rate was set to 100% when hydrothermal heat treatment was performed under high pressure, whereas, under no pressure, 80% of the filling rate was maintained. Second, the temperature of the electric furnace was set to 200 °C, and HS was performed under pressures of 0, 15, and 25 MPa for 24 h. After the HS treatment, the autoclave container was cooled down at room temperature for over 1 h. Subsequently, the samples were washed with pure water three times and filtered through a suction filter (Millipore, Merck). Finally, the refined samples were dried in an electric furnace at 90 °C for 24 h. The summarized procedure is shown in Figure 1.

### 2.2. Characterization

Changes in the surface morphology (structure, shape, and size) of the obtained samples were observed using field-emission scanning electron microscopy (FE-SEM) (JSM-7610F, Jeol Ltd., Tokyo, Japan) operated at 15 kV. To identify and visualize the elemental distributions of Ti, Nb, and O in the as-obtained samples, energy-dispersive X-ray (SEM-EDX) (JSM-7610F, Jeol Ltd.) analysis was performed at 15 kV with a cps of 3000. Nine points were selected from each sample for analytical sampling. The crystalline phases of the obtained samples were identified using X-ray diffraction (XRD) (SmartLab, Rigaku, Tokyo, Japan) with Cu–Kα radiation (40 kV, 30 mA), within the 2θ ranges of 5–90° and a scanning rate of 2θ = 20°/min. The ORR was estimated using electrochemical measurements. For the electrochemical measurements, a glassy carbon (GC) rotating disk electrode (RDE; 3.0 mm diameter) was used as the working electrode. The GC electrode was first polished with a 1.0 μm diamond paste and subsequently with a 0.05 µm alumina paste. Thereafter, 5 mg of the prepared powder samples was added to a mixture of 900 µL ethanol and 100 µL Nafion solution and dispersed ultrasonically for 40 min. Subsequently, 3 µL of the resulting slurry was dropped onto the polished GC electrode and dried at room temperature in air. All the electrochemical measurements, including cyclic voltammetry (CV) and linear sweep voltammetry (LSV), were recorded on a computerized electrochemical analyzer (704ES, CH Instruments, Inc., Bee Cave, TX, USA). A platinum coil was used as the counter electrode and Ag/AgCl (saturated KCl) electrodes were used as reference electrodes. The platinum coil was inserted into a glass tube, and an ion-permeable membrane was attached to the apical end of the glass tube. For all the electrochemical measurements, a 0.1 M KOH solution was used as the electrolyte solution. Before the measurements, we treated N_2_ or O_2_ gas for purging the solution for 20 min at a flow rate of 50 mL/min. The CV and LSV curves were measured at scan rates of 100 and 10 mV/s, respectively. Herein, from the results of LSV curves, the *n* involved in the ORR process can be calculated through the Koutecký–Levich (K–L) plots, using the following equations [32,33]:$$\frac{1}{I}=\frac{1}{I_{K}}+\frac{1}{I_{D}}=\frac{1}{I_{K}}+\frac{1}{0.62nFAC_{0}D_{O}^{2/3}\nu^{-1/6}\omega^{1/2}}$$
where *I* is the measured current density, *I*_K_ is the kinetic-limiting current density, *I*_D_ is the diffusion-limiting current density, *A* is the surface area of the electrode, *ω* is the angular velocity of the electrode, *F* is the Faraday constant (96,485 C/mol), *D_O_* is the diffusion coefficient of O_2_ in the electrolyte (1.87 × 10^−5^ cm^2^/s), *ν* is the kinematic viscosity (0.01 cm^2^/s) of the electrolyte, and *C*_0_ is the bulk concentration of O_2_ in the electrolyte (1.21 × 10^−6^ mol/cm^2^) [34]. In this study, for a comparison, a Pt/C suspension was also prepared and dropped on the RDE as a benchmark following the same procedure as described above. The LSV rotation speed was 1600 rpm, and Pt/C (20 wt% Pt loading on Vulcan XC-72) was purchased from Sigma-Aldrich.

## 3. Results and Discussion

Figure 2 illustrates the FE-SEM images of the surfaces of the obtained samples of TiO_2_ incorporated with different amounts of Nb under different load pressures. The obtained samples are termed as Nb x%-TiO_2_ in this study, where x denotes the Nb doping level. Ellipsoid morphologies were observed in all the obtained samples. Moreover, the size of the individual ellipsoid particle was approximately 200 nm, and the particle assembly was several microns. Generally, representative TiO_2_ crystalline structures, i.e., anatase and rutile, are shown as octahedral or tetrahedral structures. However, the obtained structure of Nb 10%-TiO_2_ in this study was very similar to the spherical morphology; therefore, the HS method was able to selectively synthesize the spherical morphologies, leading to an expected high-performance electrocatalytic activity due to the high surface area. In addition, with an increase in the load pressure, a trend was observed between the particles in many aspects. Therefore, it was assumed that the crystallinity of the obtained Nb 10%-TiO_2_ samples decreased with an increase in the loading pressure, as compared with that of pure TiO_2_.

Figure 3 shows the XRD profiles of the Nb 10%-TiO_2_ samples prepared under various load pressures. As can be confirmed, many anatase peaks from the TiO_2_ phase were detected. Remarkably, Nb-incorporated TiO_2_ phases such as TiNb_2_O_7_ were detected at approximately 2θ = 23°, indicating that Nb-incorporated TiO_2_ can be successfully synthesized via this process. Based on the XRD profile, when the load pressure was increased to 25 MPa, the peak intensity of the (204) plane for TiNb_2_O_7_ to the half-width increased from 6.97 to 9.99, with an expected increase in TiNb_2_O_7_ (Figure 3c). It can be assumed that the anatase phase grew selectively on increasing the load pressure up to 25 MPa. In particular, the (204) plane was selected as the representative anatase TiO_2_ phase, and the half-width and peak intensity were analyzed to determine the effect of the load pressures during the process. With an increase in the load pressure, both the half-width and peak intensity increased (Figure 3b). Considering this result, it can be assumed that the obtained amount of anatase increased, based on the increase in the peak intensity; however, the crystallinity decreased, as indicated by the decline in the half-width. The SEM results (Figure 2) support this claim. Thus, it was concluded that the load pressure has a significant effect on the modulation of the morphology.

To determine the relationship between the decline in the crystallinity of anatase and the incorporated Nb content in the obtained crystallinity, EDS analysis was conducted under different load pressures at a fixed doping level of Nb-10% (Figure 4). Ti, Nb, and O elements were detected in all the obtained samples. The Nb/Ti ratio without load pressure was 10.7; therefore, it can be assumed that the Nb doping level is almost 10%. These results suggest that there exists a direct relationship between the Nb doping level and Nb content in the obtained samples, and it is assumed that the synthesized Nb oxides are homogeneously dispersed in the obtained samples. Interestingly, the Nb/Ti ratio increased with an increase in the load pressure owing to an increase in the quantity of TiNb_2_O_7_. In the case of the Nb 10%-TiO_2_ samples treated at 15 and 25 MPa, the Nb contents in the samples were 11.2% and 18.6%, respectively, when the Nb doping level was fixed at 10% (Figure 4b). Therefore, it can be considered that the load pressures affected the control over the Nb doping level. Further details regarding the mechanism of increase in the Nb content with the increasing load pressure are presented in Figure 5. A common type of conversion reaction is the alloying or compounding of atomic species through a diffusion couple, which often results in porous structures [35,36,37]. Void formation can occur within particles when the species from the particle core diffuse outward more quickly than the inward diffusion of the reactive species, thereby causing the reaction to occur on or near the particle surface. It is a manifestation of the Kirkendall effect of void formation owing to this imbalance on the diffusion rates [38,39]. The conversion of metal into oxides often occurs through the Kirkendall effect, where the outward diffusion of metal atoms from the core is faster than the inward diffusion of the reactive species, resulting in void formation. Oxidation is a common particle conversion chemical reaction [40,41,42]. The oxidation of particles involves a process similar to the oxidation of bulk metals, where a thin oxide layer forms on the metal surface, followed by the simultaneous outward diffusion of metal ions through the oxide scale and inward diffusion of oxygen into the particles [35]. As compared to cations, metal ions often diffuse outward more quickly than oxygen diffuses inward, which is consistent with the larger ionic radius of anions. The balance in the diffusion rates defines the structure of the oxidized product. More complex structures, rather than simply incorporating oxygen in the initial structure wherever the metal is present, can emerge when the outward diffusion of the metal is significant. If the inward diffusion of the oxidizing species into the particles is faster than the outward diffusion of metal cations, then the particle shape (neglecting the change in volume to accommodate this oxygen incorporation) is typically preserved. This behavior is often detected during the initial stages of oxidation. At the intermediate stages of oxidation, core/shell structures are often observed, several examples of which are known [43,44,45]. The Kirkendall effect, where the outward diffusion of the metal is faster than the inward diffusion of oxygen, can drive substantial morphological changes during particle oxidation. In this study, Nb oxides and/or solid-solved-state Nb elements, including oxygen vacancies, diffused toward the surface during crystal growth. Therefore, the inward diffusion rate near the surface was significantly higher than the outward diffusion rate; this difference in the diffusion rates resulted in the “different oxygen vacancy level” in the structure. As the reactions continue, inward void flow occurs to balance the unequal matter flow, which eventually results in the formation of an internal void in the TiO_2_ bulk structure. Meanwhile, phase transformation occurs simultaneously with the formation of the hollow structure owing to the activated diffusion. Finally, the Nb content increases.

To determine the effect of the Nb doping level on the crystallinity of the obtained Nb-TiO_2_ samples, various samples with different Nb doping levels were synthesized, without being subjected to the loading pressures; subsequently, the synthesized morphologies were observed via SEM (Figure 6). In the case of pure TiO_2_ particles, a spherical morphology was observed (Image of Nb-0%). However, an ellipsoidal morphology was observed in the Nb-doped samples. The size of the obtained individual particles was approximately 200–300 nm, and the particle assembly, comprising ellipsoidal particles, was several microns. This phenomenon can be explained by the abovementioned hypothesis, which states that doped Nb can selectively affect the growth plane of TiO_2_; therefore, the grain growth of TiO_2_ was suppressed on increasing the Nb content.

Figure 7 shows the XRD profiles of the obtained Nb-TiO_2_ samples with different Nb doping levels. Similar to the Nb-TiO_2_ samples subjected to various load pressures, it can be confirmed that many anatase peaks from the TiO_2_ phase were detected. Similarly, Nb-incorporated TiO_2_ phases such as TiNb_2_O_7_ were detected at approximately 2θ = 23°, indicating that Nb-incorporated TiO_2_ can be successfully synthesized via this process (Figure 7a). Interestingly, the diffraction angle of the (204) plane shifted to a low angle degree as the Nb content increased, as shown in Figure 7b. This result can be explained as follows: The lattice constant of the TiO_2_ crystal increased owing to the increase in the Nb content in the TiO_2_ crystal. The exchange of Ti in the TiO_2_ crystal with Nb during the process indicates that Nb doping was successfully realized [46,47,48]. In addition, the crystallinity decreased on increasing the Nb doping level, based on the decrease in the half-width (Figure 7c).

To determine the electrocatalytic activity of the obtained Nb-TiO_2_ samples for the ORR, the ORR performance was estimated using electrochemical measurements (Figure 8). The effect of the loading pressure and Nb doping level on the electrocatalytic activity was evaluated using the Nb-TiO_2_ samples that were synthesized using different load pressures (Figure 8a) and Nb doping levels (Figure 8b). The samples were denoted as 0, 15, and 25 MPa according to the load pressure utilized and 0%, 10%, and 20% according to the doping levels used for synthesis. Overall, the ORR onset potential of all the LSV curves for different loading pressures decreased as 0.691, 0.672, and 0.624 V for 0, 15, and 25 MPa, respectively. However, for different Nb doping level series, the ORR onset potential increased until 10% and subsequently decreased until 20%. In addition, the electron transfer number *n* exhibited the maximum values at 0 MPa, slightly decreasing at 15 MPa and increasing at 25 MPa, within the error ranges. These results suggest that high load pressures induce an increase in the Nb content and TiNb_2_O_7_ phase, thereby suppressing the ORR activity. The reported Nb-doped TiO_2_ samples synthesized via the sol–gel method and the 0 MPa samples with the maximum values of *i*_ORR_ and *E*_ORR_ were compared. The value of *E*_ORR_ was very similar to the reference value (0.7 V); however, the value of *i*_ORR_ for the 0 MPa samples synthesized in this study was 10 times higher than the reference value. It can be assumed that the obtained amount of TiO_2-x_ with electron transfer properties, particularly Ti_4_O_7_ with high conductivity [24,49,50], was suppressed under this synthetic process owing to the Kirkendall effect (Figure 5); therefore, the value of *i*_ORR_ at 0 MPa decreased. The oxygen vacancies, which contribute toward the electrical transfer ability in oxides, decreased according to the pressure; therefore, it is necessary to further study this issue to perform this experiment at low temperatures. In the case of the Nb doping level series, the values of *i*_ORR_, *E*_ORR_, and *n* increased up to 10% and then decreased thereafter. This is interesting because the optimized Nb content in this process, that is, Nb 10%, is the optimal condition for achieving high electrocatalytic activity. In addition, for a comparison, a Pt/20% C electrode was used as a benchmark in this study. The value of *n* for the 10% Nb-doped sample was 3.25, whereas that for the Pt/20% C electrode was 3.93; this indicates that the samples prepared in this study were comparable to the benchmark, without employing Pt and C. This result, in turn, shows that the Pt- and C-free samples prepared in this study have the potential to enhance the ORR. In general, there are two reaction pathways for the ORR: two- and four-electron reactions. Regarding the application in energy systems, such as batteries and fuel cells, the preferential occurrence of the four-electron reaction for the ORR is important in order to improve the energy efficiency. Thus, it can be considered that the key factor determining the reaction pathway is *n*. When *n* exceeds 3.0, the four-electron reaction will be dominant in the ORR. When the Nb doping amount is 10%, the value of *n* is found to be 3.3, thus indicating that the four-electron reaction is dominant. By contrast, when the Nb doping amounts are 1% and 20%, the *n* values are below 3.0, suggesting that the two-electron reaction is dominant. These results indicate that the Nb doping level can affect the selectivity of the ORR between the two- and four-electron pathways. Furthermore, based on these results, an increase in NbTi_4_O_7_, instead of a decrease in highly conductive Ti_4_O_7_, leads to a low *n* value, which, in turn, results in low electrocatalytic activity during this process.

In summary, the relationship between the crystal structure and electrocatalytic activity can be explained as follows: First, oxygen vacancies contributing toward conductivity can be obtained via Nb doping. Second, Nb contents <10% induce an increase in TiNb_2_O_7_, leading to a decrease in conductivity. Third, a decrease in the surface area of the particles results in a decrease in the electrocatalytic effect. These summarized considerations were assembled, as shown in the following schematic (Figure 9).

In the case of Nb contents <10%, the surface area increased owing to crystal growth, and the oxygen vacancies that contribute toward conductivity increased. However, in the case of Nb contents >10%, the TiNb_2_O_7_ content increased, resulting in the formation of sintering structures; this decreased the surface area and electrocatalytic activity. In conclusion, the most optimized condition to produce high-performance catalysts is 10% Nb-incorporated Ti oxides.

## 4. Conclusions

In this study, we systematically demonstrated the relationship between the electrocatalytic activity and the load pressure and Nb content by synthesizing Nb-incorporated TiO_2_ under a hydrothermal process. In the case of the samples obtained using various load pressures during the process, the Nb content and the obtained amount of TiNb_2_O_7_ increased, leading to the synthesis of particles with ellipsoid morphologies. The obtained individual particles were approximately 200 nm. In addition, when the Nb doping amount was increased, XRD results showed that the diffraction angle shifted to a lower angle and the half-width decreased. This implies that Ti atoms were exchanged with Nb atoms during the process; therefore, the TiO_2_ crystallinity decreased. Using these samples, it was confirmed that the 10% Nb doping level was the optimal condition to achieve high electrocatalytic activity in terms of *i*_ORR_ and *E*_ORR_. This also suggests that the 10% Nb-doped samples have the ability to enhance electrocatalytic activity.

## Figures and Tables

**Figure 1 materials-15-01633-f001:**
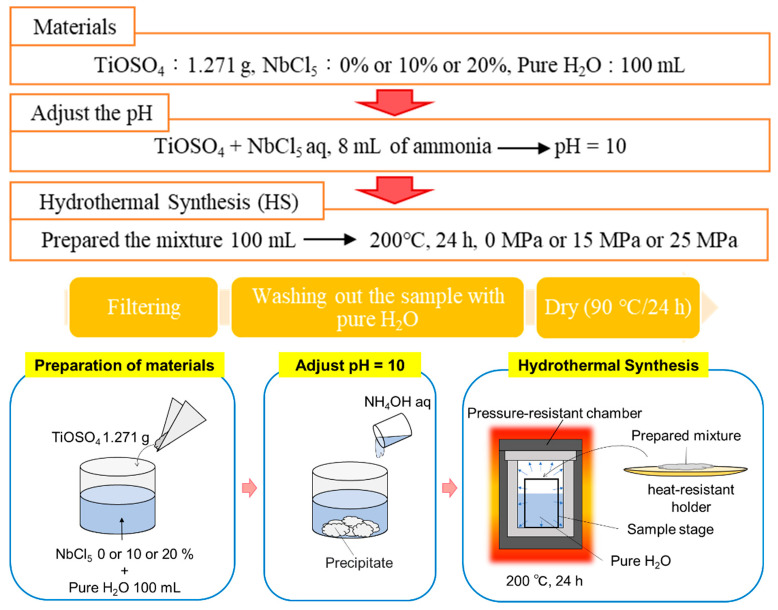
Procedure of sample preparation.

**Figure 2 materials-15-01633-f002:**
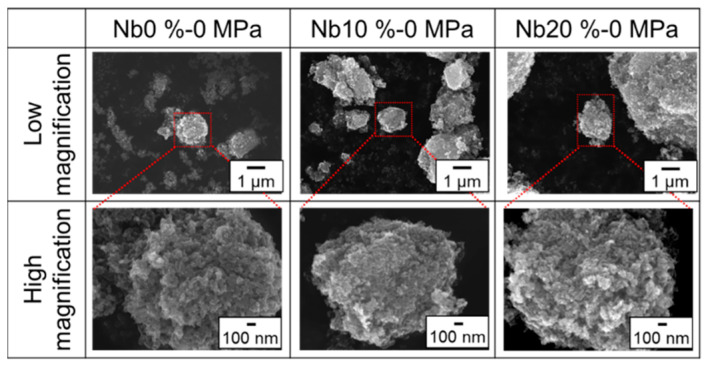
SEM images of obtained Nb-TiO_2_ samples under different load pressures. Nb concentration was fixed at 10%.

**Figure 3 materials-15-01633-f003:**
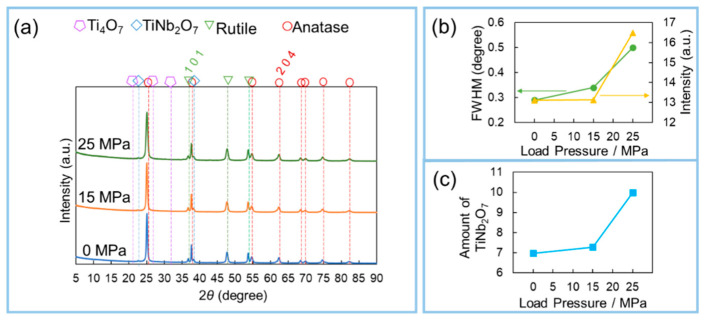
(**a**) XRD patterns of obtained Nb-TiO_2_ samples under different load pressures. Nb concentration was fixed at 10%; (**b**) Analytical results of the relationship between the half-width and peak intensity of the (204) plane; (**c**) Quantitative analysis of the amount of obtained TiNb_2_O_7_.

**Figure 4 materials-15-01633-f004:**
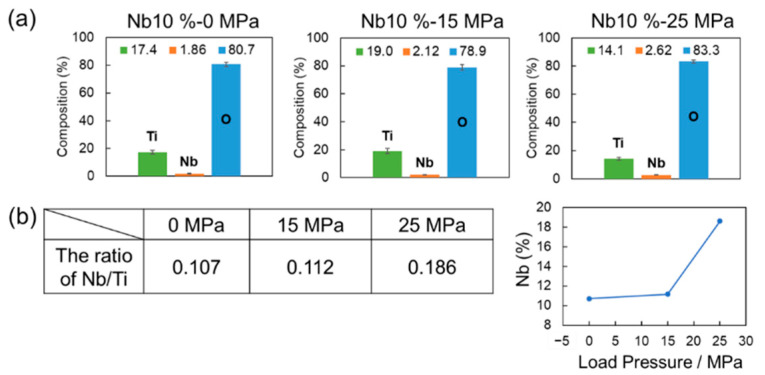
(**a**) Composition analysis of obtained Nb-TiO_2_ samples under different load pressures, as determined via EDX. Nb concentration was fixed at 10%; (**b**) Quantitative analysis of the Nb/Ti ratio and amount of doped Nb.

**Figure 5 materials-15-01633-f005:**
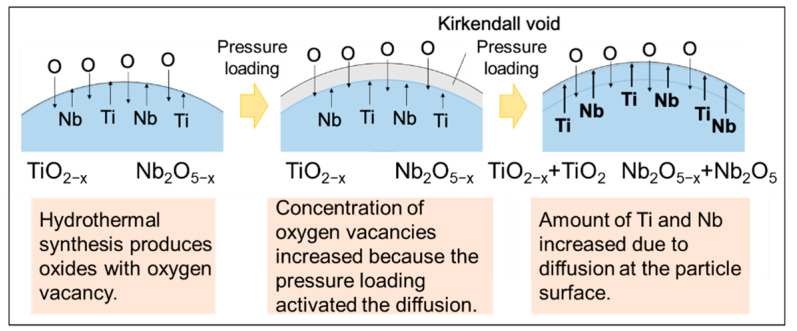
Schematic of the mechanism governing the change in the amount of incorporated Nb under applied pressures.

**Figure 6 materials-15-01633-f006:**
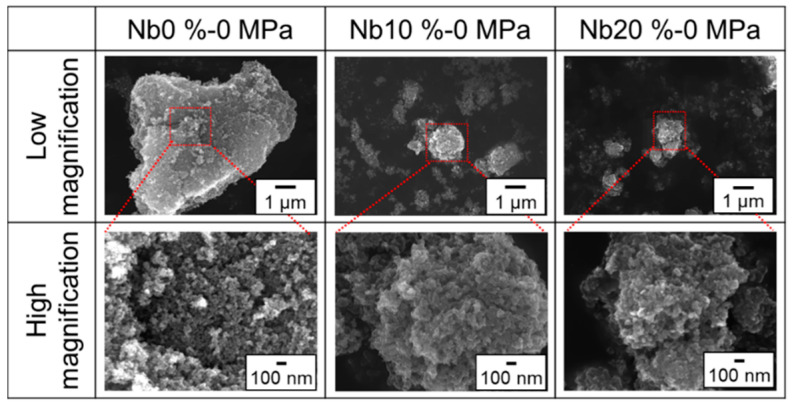
SEM images of the obtained Nb-TiO_2_ samples under different Nb doping levels without loading pressure.

**Figure 7 materials-15-01633-f007:**
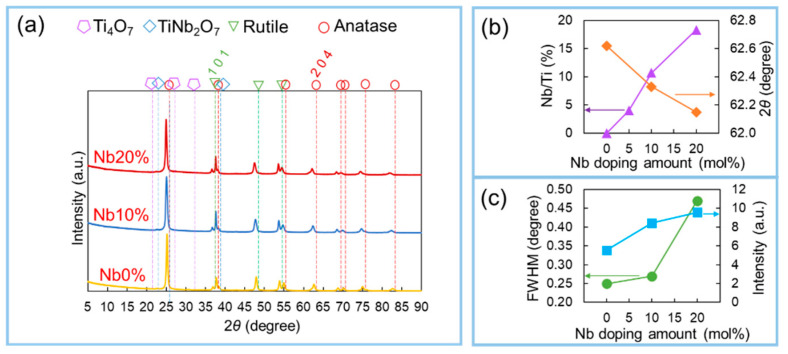
(**a**) XRD pattern of obtained Nb-TiO_2_ samples under different Nb doping levels; (**b**) Relationship between the percentage of Nb/Ti and diffraction angle of the (204) plane according to different Nb doping levels; (**c**) Analytical result of the relationship between the half-width and peak intensity of the (204) plane according to different Nb doping levels.

**Figure 8 materials-15-01633-f008:**
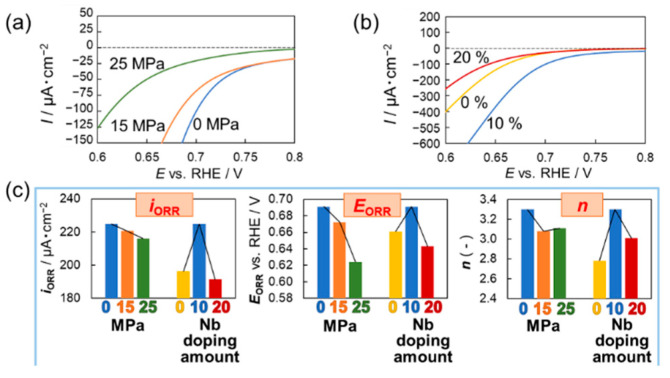
Evaluation of the electrocatalytic activity of obtained Nb-TiO_2_ samples under different (**a**) loading pressures; (**b**) Nb doping levels; (**c**) Analytical results of the trends for *i*_ORR_, *E*_ORR_, and *n* within the obtained Nb-TiO_2_ samples under different loading pressures and Nb doping levels.

**Figure 9 materials-15-01633-f009:**
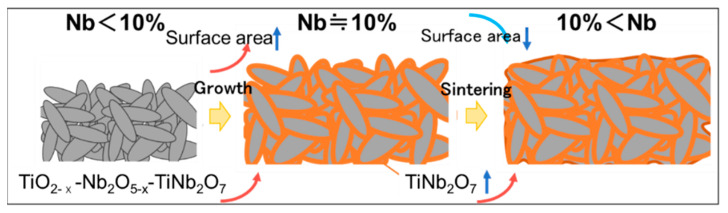
Schematic of the effect of Nb content during the process.

## Data Availability

Not applicable.

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
