# Peer review of "Systematic Study of Effective Hydrothermal Synthesis to Fabricate Nb-Incorporated TiO2 for Oxygen Reduction Reaction"

_materials, 2022, doi:10.3390/ma15051633_

Round 1

Reviewer 1 Report

In this work, Lee et al. prepared the Nb-incorporated TiO2 by hydrothermal synthesis for oxygen reduction reaction applition. This is an important work with some novelty. However, some changes should be made to further improve the quality of the manuscript.

  1. Figure 2 & Figure 6, please mark the enlarged area shown in the bottom.
  2. If possible, please add the error bar of the iorr, Eorr and n.
  3. Schematic of the effect of Nb content during the process. No explanation for Nb>10%, please double-check the scheme.
  4. No single fuel cell is assembled, and only ORR performance is studied. Therefore, I suggest the authors to change the title to "Systematic study of effective hydrothermal synthesis to fabri-2 cate Nb-incorporated TiO2 for Oxygen Reduction Reaction"

Author Response

Please check up the attached file, thank you.

Reviewer 2 Report

The authors have tried to synthesize Nb incorporated TiO2 for fuel cells, however, there are several issues with this paper.

  1. The abstract is too long for readers, please rewrite it while including important aspects.
  2. The authors have vaguely elaborated the SEM and XRD images for different synthesis approaches. No need to elaborate them, either merge or move to the supporting file.
  3. The main heart of the paper i.e., the fuel application is poorly presented. The explanation is very little and data is not up to the mark Therefore, write the application part again with proper explanations.
  4. Give at least 1 comparative table that compares your material with previous literature.
  5. I found the novelty of work very weak. So explain the novelty of paper with proper justifications.
  6. Prepare Figure 1 again. It is appearing a basic reactions figure, draw a neat synthesis flow, and provide an attractive graphical abstract.
  7. The paper is full of grammatical errors. So please check the paper with some English native speakers.
  8. There is no sense to give SEM images at the same magnifications so please reduce the SEM figures at the same scale.
  9. Compare the XRD data with their simulated counterparts.
  10. Take care of subscripts and superscripts in the whole manuscript, especially in title of paper.

Author Response

(The authors gave the same response as above.)

Reviewer 3 Report

The manuscript report with the title "Systematic Study of Effective Hydrothermal Synthesis to fabricate Nb-incorporated TiO2 for Fuel Cells" describes the hydrothermal synthesis of Nb-incorporated TiO2 and its characterization. After reading this review, it seems that the subject matter is pretty good and quite interesting to the broad spectrum of readers, even suitable for the journal. However, the scientific soundness of the article diminishes because of the incomplete characterization and lack of reasoning. Before publishing the manuscript, the authors might consider the following suggestions to improve their manuscript:

  1. How does the author recognize particle morphology as ellipsoid morphology? There was no such characteristic evidence that made clear their speculation. The author should provide clear evidence.
  2. Although the enhanced catalytic activity for 10% doping of Nb and the reduction of catalytic activity with an increased amount of Nb doping was not clear to me, the author should clarify. As a result, the author's statement as a summary is fully speculative and vague regarding the surface area. Since the surface area of a catalyst does not signify anything, especially for electrochemistry, the author should investigate the electrochemically active surface area.
  3. For better understanding and comparison, the authors should include a benchmark catalyst ORR activity, for example, pt or some other metal that is used as a benchmark.
  4. Along with the EDX composition analysis chart, the author should include EDX-spectra and elemental mapping images, which is a more acceptable way of representing EDX data.
  5. In the result and discussion section, on line 139, the number of figure "Figure 2A" might be wrong. The author should rectify this.
  6. In the introduction, the second paragraph from lines 48–58 seems to be unnecessary and should be avoided.
  7. The title should be changed. I would suggest omitting the term "fuel cell" or including some fuel cell data like single-cell performance.
  8. On page 8, lines 265-266, the used unit might be wrong, and it does not seem to be the %. It seems to be MPa.
  9. The authors mentioned, "A few research reported that the 10 % Nb-doped in TiO2 has the potential to improve the ORR, therefore this study took the information as a base condition to start the experiment." They have used only one reference. Either they should change their sentence statement or increase the number of references.

Author Response

(The authors gave the same response as above.)

Reviewer 4 Report

The manuscript "Syntematic study of effective hydrothermal synthesis to fabricate Nb-incorporated Ti)2 for fuel cells" is well written.

 The conclusion is vague. Do you Incorporation of NB was explained? View the research https://doi.org/10.3390/gels7040195. Connect with results.

  The introduction is weak. Many sentences and disorganized information, which make the flow of the text difficult. the amount of work using Nd as precursors is enormous, and no knowledge gap was exposed in the introduction. Several other works evaluate the properties of these materials, with consistent analysis and data, so it is necessary to search for a specific gap and make the need for research clearer.

  This research does no confirm the potential use of materials in different functionalities, it hasn't even been evaluated. The conclusion need to present the findings in a clear way.

This manuscript can be accepted after minor revision. 

Author Response

(The authors gave the same response as above.)

Round 2

Reviewer 2 Report

The revised manuscript can be accepted now but I wonder why the number of authors has been increased at this stage.

Author Response

Please check up the attached file.

Reviewer 3 Report

Although the authors have revised the manuscript draft and improved a lot, there are several issues that remain unresolved that should be rectified before publishing the draft, which are listed below.

  1. The first line of the abstract does not make any meaningful expression, and it is hard to follow. It should be changed with a proper replacement.
  2. In response to comment 3, the author provided the pt/C ORR activity as a data table, which is incomplete. After completing the table with I-ORR and E-ORR data, it should be included in the main draft. Along with this, the LSV curve of Pt/C for ORR activity should also be included in the graphs in Figures 8a and b.
  3. In response to comment 4, the author provided EDX data that was not trustworthy. There are lots of impurities like C, N, S, and Cl as compositional peaks. Please make sure to use trustworthy EDX data and include it in the main draft.

Author Response

Please check up the attached file.
